# Covalently bridging graphene edges for improving mechanical and electrical properties of fibers

Ling Ding[1,9], Tianqi Xu[1,2,9], Jiawen Zhang[1,2,9], Jinpeng Ji[1,9], Zhaotao Song[1,2,9], Yanan Zhang[1], Yijun Xu[3], Tong Liu[3], Yang Liu[3], Zihan Zhang[4], Wenbin Gong[5], Yunong Wang[1], Zhenzhong Shi[1], Renzhi Ma[4], Jianxin Geng[2,6], Huynh Thien Ngo[4], Fengxia Geng[1,7] ✉ & Zhongfan Liu[7,8]

Assembling graphene sheets into macroscopic fibers with graphitic layers uniaxially aligned along the fiber axis is of both fundamental and technological importance. However, the optimal performance of graphene-based fibers has been far lower than what is expected based on the properties of individual graphene. Here we show that both mechanical properties and electrical conductivity of graphene-based fibers can be significantly improved if bridges are created between graphene edges through covalent conjugating aromatic amide bonds. The improved electrical conductivity is likely due to extended electron conjugation over the aromatic amide bridged graphene sheets. The larger sheets also result in improved π-π stacking, which, along with the robust aromatic amide linkage, provides high mechanical strength. In our experiments, graphene edges were bridged using the established wet-spinning technique in the presence of an aromatic amine linker, which selectively reacts to carboxyl groups at the graphene edge sites. This technique is already industrial and can be easily upscaled. Our methodology thus paves the way to the fabrication of high-performance macroscopic graphene fibers under optimal techno-economic and ecological conditions.

Carbon fibers combine light weight with ultra-high mechanical strength and are indispensable components in many high-end applications, such as aerospace/aviation industry, civil engineering, and competitive sports[1–4]. The basic structural unit of a carbon fiber at the atomic level is a graphene layer with carbon atoms in a hexagonal lattice arrangement, and these sheets are essentially aligned parallel to the long axis of the fiber[3,4]. The carbon atoms in such a unit are linked by strong covalent σ bonds with a strength of 400 kJ mol$^{-1}$, which

imparts a high theoretical fracture strength of 130 GPa and a Young's modulus of 1.0 TPa[5,6]. Commercially available polyacrylonitrile and mesophase pitch-based carbon fibers are produced by fusion of organic fragments followed by extreme high-temperature annealing to achieve graphitization. However, the formed fibers have only small graphene units (a few to tens of nanometers) and low-order turbostratic stacking. In contrast, mechanical or chemical exfoliation from graphite can produce graphene with sizes of the order of tens of

[1]College of Energy; School of Physical Science and Technology & Institute for Advanced Study, Soochow University, Suzhou 215006, China. [2]Beijing University of Chemical Technology, 100029 Beijing, China. [3]Suzhou Institute of Nano-Tech and Nano-Bionics, Chinese Academy of Sciences, Suzhou 215123, China. [4]National Institute for Materials Science, Tsukuba, Ibaraki 305-0044, Japan. [5]School of Physics and Energy, Xuzhou University of Technology, Xuzhou, China. [6]State Key Laboratory of Separation Membranes and Membrane Processes; School of Material Science and Engineering, Tiangong University, Tianjin 300387, China. [7]Beijing Graphene Institute, 100095 Beijing, China. [8]Peking University, 100871 Beijing, China. [9]These authors contributed equally: Ling Ding, Tianqi Xu, Jiawen Zhang, Jinpeng Ji, Zhaotao Song. ✉e-mail: gengfx@suda.edu.cn

micrometers, the direct assembly of which is believed to be an effective alternative approach for next-generation high-performance carbon fibers[7–9]. The large size of the units also offer the possibility to improve the electron conductivity of the assembly. Graphene-based carbon fibers thus combine light weight, high strength, and excellent electrical conductivity, and are therefore of great interest as potential low cost, light weight substitutes for copper wires in weight sensitive applications, such as large military and scientific satellites.

Macroscopic graphene fibers have been prepared from two-dimensional graphene oxide (GO) sheets through a solution-based wet-spinning technology followed by treating them under reducing conditions[10]. These reports have highlighted the importance of regular alignment of graphene sheets and decreasing structural defects to enhance the mechanical and electrical properties of graphene fibers[11–13]. The first report estimated the tensile strength of chemically reduced GO fibers to be around 200 MPa[10] and recently a benchmark value of 2.25 GPa was reached by flattening random graphene wrinkles[13]. Annealing at very high temperatures (>2500 °C) was found to annihilate atomic defects on graphene sheets and promote the formation of graphitic crystallites[11,14], resulting in graphene fibers with a tensile strength of 3.4 GPa and Young's modulus of 342 GPa[13]. However, using high annealing temperatures is often not desirable from both economic and ecological perspectives. Thus, if we want to replicate the excellent physicochemical properties of graphene in macroscale graphene fibers, it is of utmost importance to develop new strategies to prepare high performance graphene fibers, preferably at near room temperature.

Given that previous efforts to align sheets along the fiber axis have met with some success, the integrated tensile fracture stress as well as electrical conductivity are predominantly determined by in-plane inter-sheet interactions at the sheet edges. In graphene fibers reported hitherto, individual graphene sheets are connected mainly in the overlapping stacking direction through π-π bonding (and electrostatic interactions) with little or no in-plane inter-sheet junctions along the fiber axis[7–14], which places an upper limit on the mechanical properties of the macroscopic graphene fibers formed therefrom. Thus, the key to translating the excellent properties of individual graphene to the macroscopic scale lies in preventing tensile fracture and in providing electron conducting pathways, particularly at sheet edges.

In this context, we report a simple strategy to create bridges at the edges of individual graphene sheets through covalent conjugation of an aromatic amide bond, by selectively reacting the carboxyl groups at the sheet edges with an aromatic amine. Results show that a significant improvement in both mechanical and electrical conducting performance can be achieved for the assembly. The design motif and structural model of the resultant graphene fibers with amide linkages are shown in Fig. 1a, b. If we consider an amide linkage, it is known that the conjugation of the N lone pair with the carbonyl group in amide (−C(=O)NH−) generates a three-center π-system in a rigidly planar configuration[15]. The aromatic amide that bridges the graphene sheets thus can serve to extend electron conjugation over larger sheet areas. Such links offer three important advantages. First, there is a possibility to create extended conjugation between the aromatic amide bridges and the neighboring graphene sheets. Larger linked sheets are expected to show enhanced π–π interaction, which facilitates ordered stacking of the sheets. Secondly, the extended conjugation can form a π electron cloud over relatively larger connected areas of graphene and increase electron mobility across the graphene sheets. Finally, the robust covalent amide −C(=O)NH− bond has a high bond energy close to that of a C–C bond[15], which, along with enhanced π–π interactions, could lead to improved axial stress transfer among the partially bridged graphene sheets and thereby enhance the mechanical performance of the assembly.

## Results and discussion
### Graphene fiber assembly from GO sheets
GO sheets were prepared according to a previously reported modified Hummers method[16]. GO sheets have abundant polar oxygen-

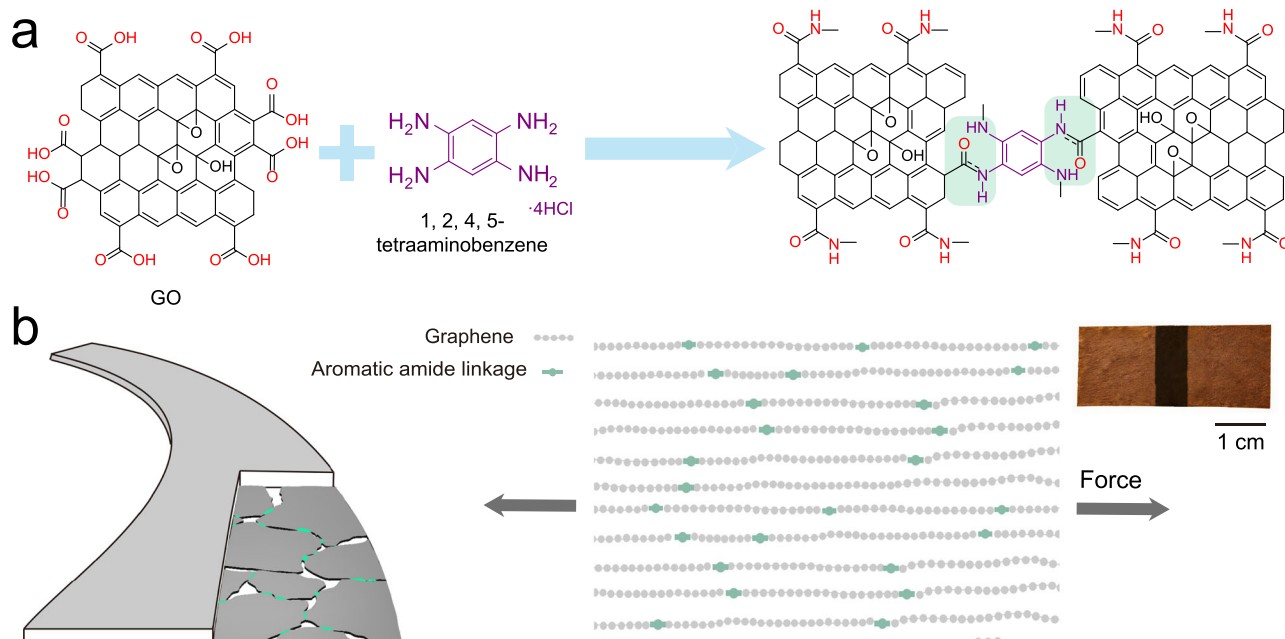

**Fig. 1 | Edge bridging of graphene sheets with covalent conjugation of aromatic amide. a** Schematic showing the principle of the assembly process. In-plane aromatic amide linkages are generated by the selective reaction of the edge carboxyl groups of graphene oxide (GO) sheets and aromatic amine, highlighted by the green shading region. GO and acid catalyze the direct amidation.
**b** Fiber constituted by regularly stacked aromatic amide-bridged graphene sheets. Inset: Digital image of two separate films with edge-connected by immersing in the aqueous coagulant amine; the connected edge transforms to a dark color. The aromatic amide linkage yields extended conjugation and enhanced π-π interaction, which, along with the inherent covalent feature of the linkage, improves both structural robustness and electrical conductivity.

containing functional groups decorating the basal plane and edges. This leads to negative surface charges and allows stable aqueous dispersions to be formed. The oxygen-containing functional groups are typically hydroxyl (C–OH), epoxy (C–O–C), and carboxyl (–C(=O)OH) groups. Atomic force microscopy (AFM) and scanning electron microscopy (SEM) (Supplementary Figs. 1, 2) showed that the GO sheets had lateral sizes predominantly in the range of 10–70 μm and a mean thickness of ~1 nm. This high aspect ratio enabled the formation of a liquid crystalline phase with short-range sheet alignment (Supplementary Fig. 3). The presence of oxygen-containing groups was verified using X-ray photoelectron spectroscopy (XPS) and the C:O atomic ratio was quantitatively estimated from the survey data to be 1.89 (Supplementary Fig. 4).

In accordance with the commonly accepted Lerf–Klinowski structural model[17], oxidized GO has hydroxyl (C–OH) and epoxide (C–O–C) groups on the basal ring planes, whereas the sheet edges are preferentially terminated with carbonyls (C=O) and carboxyl (–C(=O) OH) groups. Carboxyl groups are strong electrophiles and can react with nucleophilic amines to form amide linkages[15,18]. It has also been shown that the presence of GO or an acid enhances the electrophilicity of the carboxyl group, due to which a direct amidation is enabled without the need for an intermediate acyl substitution step[19,20]. Taking a cue from these previous works, we selected aqueous 1,2,4,5-tetra-aminobenzene tetrahydrochloride as the coagulant during fiber wet-spinning for a direct on-site condensation amidation of carboxyl groups present at GO edges. The details for the reaction mechanism are provided in Supplementary Figs. 5–7. Previous works have used simple aliphatic or aromatic diamines as cross-linkers[21–24], but it was found that in the absence of an acid these amines usually annihilated surface oxygen groups on GO, accompanied by the formation of a very high density of vacancy defects[25]. As a result, the integrated mechanical properties of the GO assembly deteriorated (Supplementary Fig. 8), and such composite membranes were mostly studied in the context of separation applications. We have considered other possible competing reactions, for example, electrostatic interaction to form a salt, reaction between carboxyl and hydroxyl to form esters (a recent report of smart way to self-crosslink GO)[26]. These reactions, if they occur, should be minor (please see Supplementary Fig. 8 for a detailed discussion). In the present case, the high H⁺ concentration ensured that the reaction of edge carboxyl groups with the Ar-NH₂ to form the amide was predominant. The other groups present on GO have relatively lower reactivities at ambient temperature, as experimentally proven from XPS characterization (Supplementary Fig. 10). It is important to note that although the amidation may not go to 100% completion and only a partial bridging of edges is achieved, its presence is expected to increase π-π interactions between larger sheets, thus providing pathways for both electron and stress transfer.

The feasibility of the direct amidation to create local amide bridges in GO was experimentally tested by connecting two individual GO films in aqueous 1,2,4,5-tetraaminobenzene tetrahydrochloride. We found that the two films merged at the edges into one larger film. The formation of a chemical connection by the conjugating amide structure darkened the sample color (digital image shown in inset of Fig. 1b and the amidation process in Supplementary Fig. 11), strongly indicating increased electron conjugation in the structure. We note here that the so-connected film showed improvement of tensile strength from ~200 to 470 MPa. Interestingly, immersing the whole film in the coagulant led to a dramatic increase in tensile strength to ~1000 MPa and also in electrical conductivity from $0.2 \times 10^5$ to $1.0 \times 10^5$ S m⁻¹. Owing to the simplicity of this reaction, we believe that it can be easily integrated with the industrially scalable wet-spinning protocol for making fibers.

Aqueous GO was made to flow through a confined channel into the coagulant of aqueous 1,2,4,5-tetraaminobenzene tetrahydrochloride. Solidification of stable fibers occurred due to the condensation reaction between the carboxyl groups at the edges of the GO sheets and the amine. We note here that no coagulation occurred when using an aromatic diamine with amino groups on the same side of the benzene ring, for example, 1,2-diaminobenzene hydrochloride, while also forming an amide bond (Supplementary Fig. 12), which confirms the presence of sheet bridging by the groups at para positions in our fiber system. The coagulated GO was chemically reduced in a second step to remove in-plane oxygen-related moieties, producing graphene fibers by ordered face-to-face stacking of graphene sheets with aromatic amide bridge connections (experimental details in Supplementary Information).

The graphene fibers obtained by our method showed a very high mechanical performance with a tensile strength of 3.54 ± 0.25 GPa and a Young's modulus of 340 ± 32 GPa. This is much higher than the best recorded values of 2.25 GPa and ~180 GPa reported so far[13]. Moreover, the electrical conductivity along the fiber axis was measured to be $1.5 \times 10^5$ S m⁻¹ (experimental details on mechanical and electrical property measurement are given in the SI), which is one order of magnitude higher than that reported for graphene fibers obtained at similar temperatures. Hence, our simple strategy is efficient in integrating the properties of individual graphene and is promising for the production of high-performance macroscopic assemblies.

## Structural characterization of graphene fibers

We noted that the fibers formed as described above had an unusual belt-type morphology, although the nozzle was circular. Figure 2a shows a schematic illustrating the possible coagulation mechanism as compared to the most prevalent solvent exchange mechanism. Typically, solvent exchange between GO dispersed in dimethylformamide and the commonly used precipitation agent ethyl acetate replicates the geometry of the confined channel, likely due to rapid solidification and the absence of constraints among the sheets. Thus, a tubular channel invariably produces a fiber with a circular cross section and conserves the random alignment of the aqueous dispersion, as confirmed from previous reports[12,13]. We also confirmed this from the morphology and microstructure characterization of our own control sample (Supplementary Figs. 13–15). An ordered assembly of GO into a compact anisotropic flat belt-like structure could only be achieved by modifying the geometry of the confinement channel using a flat channel with low height[12]. When using our selected aromatic amine as the coagulant, we found that a belt morphology with ordered assembly could be spontaneously generated. We have excluded the possible effect of a change in surface hydrophilicity after assembly to account for the unusual belt morphology (Supplementary Fig. 16). By tracking the solidification of the GO colloid at the nozzle at the start of coagulation, we found that the freshly produced fiber replicates the confinement geometry, but a flattening of the assembled fiber occurs soon after contact with the reaction with the coagulant (Supplementary Fig. 17). Thus, a fiber with belt morphology could be obtained even when using a triangular nozzle (Supplementary Fig. 18). Besides channel geometry, the assembled belt shape was found to be insensitive also to nozzle size and GO concentration. It should be noted that a larger diameter of nozzle translates to less constraint on the sheets, and the fiber structure would reflect this intrinsic interaction between sheets. However, the belt morphology was conserved for all tubular channels with inner diameters in a test range of 160–1500 μm (Supplementary Fig. 19). As mentioned above, varying the GO concentration did not change the belt-shape of the fibers, but it was possible to tune the thickness by having different GO concentrations (Supplementary Fig. 20). All these results together indicate that the absence of constraint from the confining nozzle and the intrinsic edge connection led to the unusual belt morphology.

The unusual belt morphology is indicative of an oriented stacking of sheets. This was confirmed also from scanning electron microscopy (SEM) observations; a regular stacking of sheets was observed,

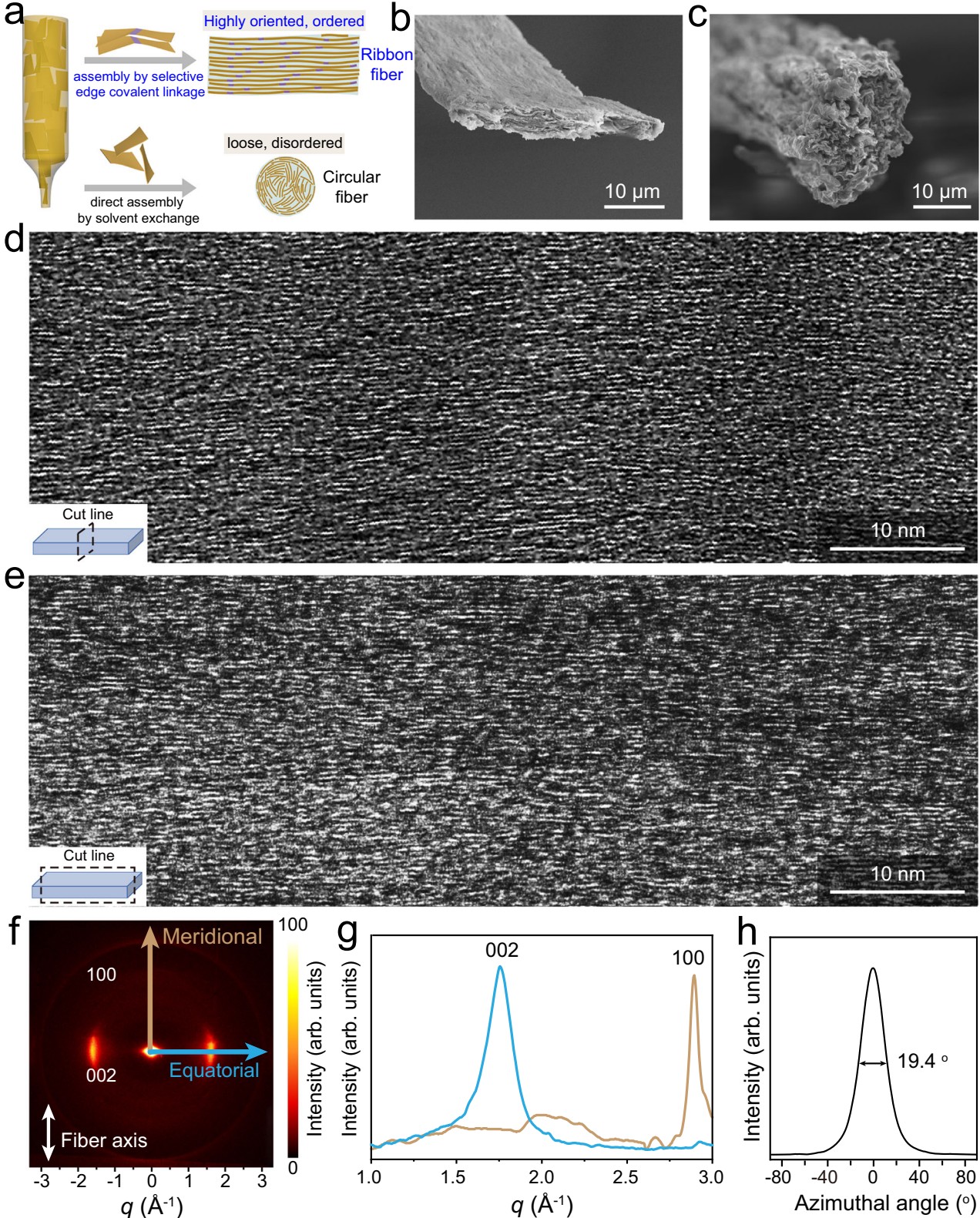

**Fig. 2 | Oriented and compact stacking of planar graphene in the assembled fiber. a** Schematic of the process to assemble GO sheets into a macroscopic fiber with belt-type morphology with highly oriented and ordered stacking structure by selective edge linkage. For comparison, the disordered assembly of GO sheets into a fiber with the usually observed circular cross-section is also illustrated, which replicates the needle geometry due to absence of constraint. Typical cross-sectional SEM images of (**b**) fiber with amide bridges and (**c**) the control fiber. Representative HR-TEM images for (**d**) a cross-sectional slice and (**e**) a transverse-sectional slice of our amide-connected fiber. **f** Two-dimensional WAXS diffraction pattern of graphene fibers. The equatorial and meridional scattering directions are indicated by cyan and bronze arrows. Color bar refer to the intensity in arbitrary units. **g** The corresponding one-dimensional scattering profile by integrating equatorial and meridional axes of (**f**, **h**) azimuthal scan integral curve for the (002) peak.

continuously extending along the longitudinal direction (Supplementary Fig. 21). The good alignment of the sheets within the macroscopic fibers along the fiber axis was further verified by polarized optical microscopy and wide-angle X-ray scattering (WAXS) pattern analysis (Supplementary Figs. 22 and 23). The improvement in ordered stacking was also supported by the reduced full width at half maximum of the basal reflection in X-ray diffraction (XRD) pattern as compared to the neat GO fiber. The interlayer spacing of the stacked sheets was determined to be 0.896 nm (Supplementary Fig. 24), which is a typical value for neat GO sheet stacking[12,13], implying the near absence of guest molecules between sheets.

The belt morphology and highly aligned structure were well conserved during the reduction step (Fig. 2b and Supplementary Figs. 25–27). The control fiber solidified by solvent exchange also maintained its circular geometry with curved and random alignment of sheets (Fig. 2c). Scanning transmission electron microscopy in SEM (STEM-in-SEM) characterization of a thin slice of the cross-section showed high alignment and regular stacking of sheets over large areas (>micrometer size) in the fiber with amide linkages (Supplementary Fig. 28), which could account for the obtained belt morphology. It should be mentioned that external stretching is applied only along the axial direction and the cross-sectional microstructure is entirely determined by intrinsic interaction between sheets, due to which some curved structures are also spotted. A longitudinal thin slice with the length of ~10 μm was also imaged to gain information on the alignment and stacking of sheets along the fiber axis direction. Along its whole length, the fiber showed straight lattice fringes (Supplementary Fig. 29). In contrast, the control fiber showed an obviously less ordered structure. Cross-sectional images showed that the sheets are curved and randomly aligned due to the lack of constraints among curvy graphene sheets. This likely accounts for its circular geometry. In the longitudinal thin slice, a local alignment of sheets is seen due to the stretching applied along the fiber axis, but misalignment of sheets was still obviously observed over large area (Fig. 2c and Supplementary Figs. 14 and 15). High-resolution TEM was employed to obtain atomic-scale information; example images for cross- and transverse sections are displayed, respectively, in Fig. 2d, e (more images in Supplementary Figs. 28 and 29). Although some stacking faults and a few random wrinkles were observed, large areas of almost straight lattice fringes in good alignment were seen, indicative of a well-aligned stacking of graphene sheets. The distance between the observed straight fringes is mostly 0.337 nm, although at stacking faults, the distances are slightly larger. The spacing was smaller than that of the control sample (~0.350 nm) and close to the stacking distance in an ideal graphite structure. The small value of the interlamellar distance is a further confirmation of the absence of guest molecules between the stacking sheets and compact stacking. The SAED pattern of the in-plane view showed the characteristic planes of the graphene structure (Supplementary Fig. 30).

XRD and WAXS analyses were performed to provide complementary information on the fiber structure. XRD data showed peaks related with compact sheet stacking and in-plane hexagonal structure of graphene (Supplementary Fig. 31). In the two-dimensional WAXS pattern, signals corresponding to (002) and (100) of graphite structure were also clearly observed (Fig. 2f, g). The full width at half maximum along the azimuthal intensity distribution of (002) (denoted as the orientation angle and is usually a measure of the degree of texture), was as small as 19.4° (Fig. 2h), indicating a heavily textured structure with a high degree of preferential alignment of the basal graphene planes. The corresponding orientation factor ($f$) was estimated to be as high as 0.896. The gravimetric density of our graphene fibers measured using the sink-float method was found to be 1.90 g cm$^{-3}$. This density is much higher than that those reported for chemically reduced graphene oxide fibers (Supplementary Table 1), the density of which are typically below 1.50 g cm$^{-3}$[11,13,27–32]. All the data thus confirm that our belt-shaped graphene fiber consists of regularly stacked planar graphene sheets with high compactness. We believe that these attributes originate from the enhanced π-π interaction among the larger amide-linked graphene sheets, which is a direct consequence of the selective condensation reaction of some carboxyl groups at the sheet edges with the aromatic amine.

The presence of amide linkages was verified by characterizing the GO fiber with a multitude of spectroscopic techniques. In $^{13}$C cross-polarization solid-state nuclear magnetic resonance (NMR) spectrum, a clear signal was observed at the chemical shift of ~150 ppm related to the amide group (Fig. 3a and Supplementary Fig. 32). Furthermore, the hydroxamic test, which is a straightforward chemical test for the formation of amide[33], gave positive results (Supplementary Fig. 33). In the XPS C 1s spectrum, an obvious shift of the signal corresponding to carboxyl to lower energy was seen (Fig. 3b), indicating the transformation of carboxyl to amide. The smaller electronegativity and the inductive effect of N relative to O atoms caused the shift. At the same time, different from the coagulation in simple amines, no intensity attenuation was observed for the −C-OH/C-O-C signal, confirming that the surface groups remained intact and the amine selectively reacted with the edge carboxyl. Figure 3c depicts the N 1s spectrum. As the difference in chemical shifts in N 1s is generally small, the data for fibers produced with an excess of aromatic amine and for the aliphatic ammonium salt are also shown. Higher binding energy values are usually related to partially positively charged N, and the signals from lower to higher energies could successively be attributed to amino, amide, and protonated amines. Using this assignment, we confirmed that the majority of N in the best amide fiber was in amide form along with minor quantities of unreacted amine. As the edge-to-edge connection only needs the reaction of 2 amines on opposite sides of the aromatic tetraamine, the majority amide signal is strongly indicative of a significant degree of boundary-to-boundary bridging of graphene sheets.

To confirm that the amide bridging is present mostly within the plane while π-π interactions hold the fiber together in the stacking direction, we investigated the stability of GO fibers both in the axial and cross-sectional stacking directions in NMP. NMP, being a good dispersant of GO, can efficiently break π-π bonds and disintegrate simply stacked GO sheets. As a result, the control fiber prepared by solvent exchange showed swelling in both cross-sectional and axial directions and the structure collapse was confirmed by a total absence of an XRD signal (Supplementary Fig. 34), which is consistent with a previous report[13]. Differently, the amide connected fiber showed anisotropic swelling, which only occurred in the stacking direction with negligible change in the fiber axis direction (Fig. 3d). It is also noted that the swelling was quite slow. No swelling was observed until after 3 h, but thereafter, the swelling continued until beyond 72 h. The structural integrity and uniform swelling were confirmed from XRD which gave a single expanded spacing of 1.424 nm. The GO surface is hydrophilic due to the rich presence of oxygen-containing groups, which means that water can also break π-π bonds by the indefinite swelling of the GO assembly. Acid in water can additionally break ion bridges formed through electrostatic attractions and was used as a preliminary test in the selection of an appropriate coagulant (Supplementary Figs. 8 and 9). The swelling behavior of our fiber in water and aqueous acid are similar to what observed for NMP, just with longer time (~120 h) and lower swelling degree (0.896–1.280 nm). The good stability in the lateral direction confirms a high-degree of strong chemical cross-linking, while the longer time needed to initiate swelling in the vertical direction could be attributed to the enhanced π-π interactions originating from the edge bridging or very low degree of chemical cross-linking.

The swollen fiber in NMP could be further delaminated into a colloid through mechanical shearing. Size analysis of the colloid by laser diffraction spectroscopy showed obvious increase in sheet size

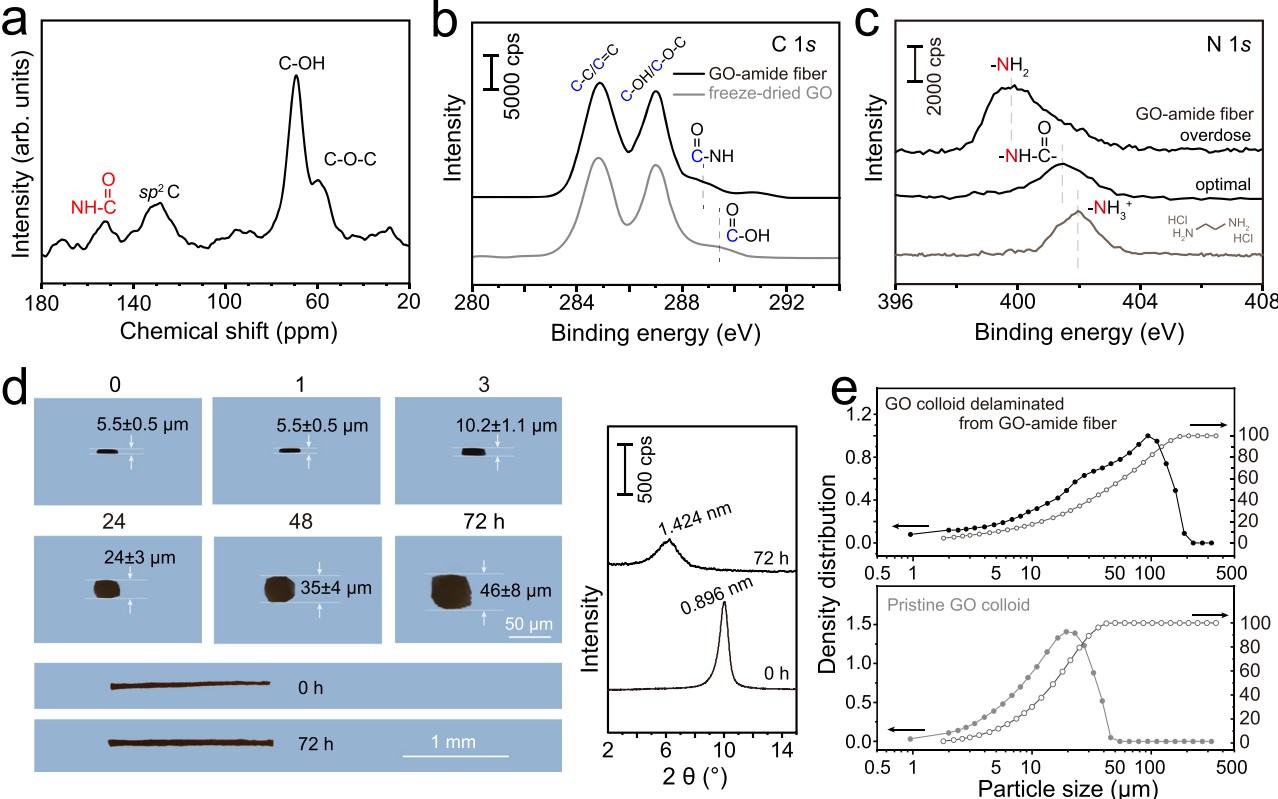

**Fig. 3 | Covalent linking of graphene edges by aromatic amide. a** Solid-state $^{13}$C cross-polarization NMR spectrum of the amide-connected GO fiber. **b** XPS C 1$s$ spectra of freeze-dried GO and the amide-connected GO fiber, (**c**) N 1$s$ spectrum of the optimal amide-connected GO fiber. To label the type of N, data for fibers fabricated with excess of amine and aliphatic ammonium salt are also shown. **d** Optical images of the amide-connected GO fiber swelling in NMP in the axial and

stacking directions; right panel: XRD patterns of the fibers. The anisotropic swelling and the longer time confirm the presence of selective amide linkages at edges and enhanced π-π interactions. Error bars correspond to the statistical error from independent measurements from at least 15 locations. **e** Sheet size distribution comparison from laser diffraction measurement of the colloid of delaminated sheets from the swollen amide-bridged GO fiber and pristine GO.

(-20–100 μm), and SEM images also showed broken parts of bridged sheets (Fig. 3e and Supplementary Fig. 35). The successful delamination confirms that weak interactions are present in the vertical stacking direction, which are mostly π-π or minimal chemical interactions. The excellent stability in the lateral direction and the obviously enlarged sheet size even after mechanical shaking indicates the presence of strong chemical interactions at edges. Taken together, these results, including the selective swelling limited to the stacking direction either in water, acid or NMP, the delamination of interlayer connections, the enlarged sheet size after delamination, along with the almost no change in the in-plane surface structure, substantiate the conclusion that the covalent amide linkage bridging is basically at the edges. In the vertical stacking direction, enhanced π-π interactions/very low-degree of chemical cross-linking is present.

### Mechanical properties of graphene fibers

While individual graphene sheets are among the strongest known materials with exceptional tensile strength and Young's modulus, the mechanical tensile behavior of macroscopic fibers consisting of assembled graphene sheets is dominated by inter-sheet interactions, especially those along the fiber axis. The presence of the covalent aromatic amide connectivity at the edges, together with strong π-π interactions from the ordered and compact packing of connected sheets, is expected to enhance the mechanical properties of the assembly. Accordingly, the amide connected GO fiber showed a tensile strength of $2.01 \pm 0.13$ GPa, and Young's modulus of $163 \pm 18$ GPa at the fracture strain of $1.25 \pm 0.10\%$ (Supplementary Fig. 36). Chemical reduction to partially restore $sp^2$ hybridization yielded graphene fibers

with increased tensile strength to $3.54 \pm 0.25$ GPa and Young's modulus of $340 \pm 32$ GPa at the fracture strain of $1.10 \pm 0.09\%$ (Fig. 4a).

These values for GO and graphene fibers are both the highest reported so far. The mechanical strength of our fiber with aromatic amide links is 1.5–1.8 times higher that of the documented best performance fiber prepared near room temperature, which has no amide connection (tensile strength of 2.25 GPa and Young's modulus of -180 GPa[13]; tensile strength of $1.9 \pm 0.1$ GPa and Young's modulus $127 \pm 24$ GPa using our GO sheets and experimental set-up). In fact, the strength value is even higher than that of graphene fibers graphitized at very high temperatures (tensile strength of 3.40 GPa and Young's modulus of -342 GPa)[13], and 1.7 times that of graphitization-annealed graphene fibers with belt morphology obtained by microfluidic assembly (1.90 GPa and Young's modulus of -309 GPa)[12]. A detailed comparison of the performance metrics of our optimal graphene fiber with values reported in the literature is presented in Fig. 4b and Supplementary Table S2. The significant enhancement in mechanical behavior could be due to the optimized inter-sheet junctions at the atomic scale and compact stacking at the microscale, both owing to our proposed assembly protocol.

Our results show that the introduction of amide links, even when present in low concentration (due to the low concentration of the coagulant or low GO oxidation degree), still has a positive effect on the mechanical performance, when compared to fibers purely solidified by solvent exchange. However, overdose of coagulant or a very high GO oxidation degree adversely affects mechanical performance; the former reduces the edge-to-edge connection and the latter deteriorates the quality of graphene sheets due to excess −OH/C-O-C

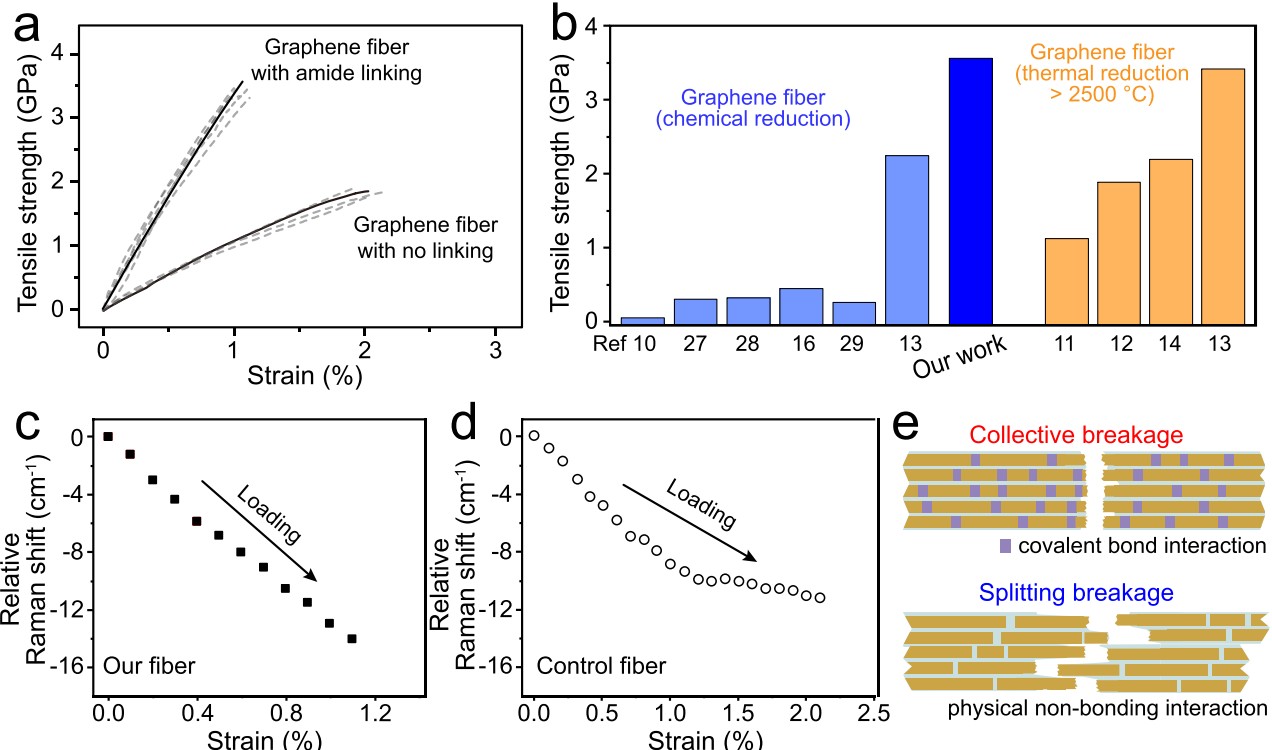

**Fig. 4 | Mechanical properties of aromatic amide-connected graphene fiber.**
**a** Typical stress-strain curves of graphene fibers after chemical reduction with and without amide linking; dotted lines are replica data. **b** Comparison of the mechanical strength with reported values in the literature. Dependence of the downshift in the Raman G-band frequency on external loading strain of (**c**) amide-bridged graphene fiber and (**d**) control fiber with no linking. **e** Collective breakage mechanism for amide-connected graphene fiber and the split breakage mechanism for the control fiber with purely physical non-bonding interactions.

(Supplementary Figs. 37 and 38). The properties of the control fiber prepared by solvent exchange are almost independent of oxidation degree due to the absence of linkers. Exclusively large sheets may also produce incomplete patching due to the rough edges of the GO sheets and steric effects (Supplementary Fig. 39). These results confirm the effectiveness of in-plane bridging of graphene and the fiber prepared under optimal conditions of amidation showed the greatest improvement in mechanical and electrical properties.

## Electrical properties of graphene fibers

The electrical conductivity of graphene-based materials is heavily influenced by structural imperfections. The high electrical conductivity results from the delocalized π-bond in the $sp^2$ hybridized graphene sheet. Heterogeneous structures, including functional groups, $sp^3$ bonds, and edges have been found to disrupt the $sp^2$ hybridized conjugated system and are therefore detrimental to electron transport[34]. In our fibers, in addition to improving the mechanical strength, the aromatic amide linkages where present also give rise to extended conjugated system with graphene, enabling improved electron conduction at the sheet edges and increasing the overall electrical conductivity. As a result, our macroscopic graphene fibers fabricated at near-room temperature achieved a notable electrical conductivity of $1.50(\pm 0.05) \times 10^5$ S m$^{-1}$. This electrical conductivity is one order of magnitude higher than that of the control fiber with no linker $(0.32(\pm 0.03) \times 10^5$ S m$^{-1})$ and previously reported graphene fibers prepared at low temperatures (Supplementary Table 3)[11–14,27–32]. The origin of the high electrical conductivity in aromatic amide-bridged samples was revealed from Hall effect measurements on film samples, which showed obvious increase in carrier concentration in the sample where aromatic amide bridges were present (Supplementary Fig. 40). These results illustrate the importance of such inter-sheet bridges in improving electron transport between graphene sheets.

With the bridging at edge and improved π-π interactions between stacking sheets, the electrical conductivity of the obtained graphene fiber is more than one order of magnitude higher than that for poly-acrylonitrile (PAN) based carbon fibers $(0.1–1.4 \times 10^5$ S m$^{-1})$. Incidentally, this value is comparable to that of mesophase pitch (MPP)-based carbon fibers (the best value of $8.3 \times 10^5$ S m$^{-1})$, without the need for extreme high-temperature annealing[14]. Although the mechanical properties are still inferior to the benchmark of carbon fibers, the simultaneous enhancement in tensile strength, modulus, and electrical conductivity of our fiber produced near room temperature highlights the importance of controlling graphene sheet assembly and the advantage of using graphene as the precursor for the fabrication of high-performance fibers.

## Stress transfer in graphene fibers

In practice, the mechanical properties of a macroscopic fiber assembly are highly dependent on the mechanism of stress transfer among individual graphene sheets. Raman spectroscopy is a useful tool to quantitatively observe this effect by measuring the shift in characteristic bands under strain[35]. As depicted in the Raman spectra (Fig. 4c and Supplementary Fig. 41), a downshift in the graphene G-band was observed for our fiber because of the stretching strain on individual sheets. This downshift increased over the entire strain range before fracture, implying that the external mechanical tensile strain was continuously transferred to graphene. In sharp contrast, for the control fiber sample solidified by solvent exchange, increasing stress transfer only occurred at low strain values (Fig. 4d and Supplementary Fig. 41). A further increase in external strain did not bring about a further G band shift, indicating that the assembly stopped working as an ensemble and internal fracture occurred. The integrity of our optimal fiber under fracture can be attributed to covalent amide connectivity among sheets at edges in addition to strong π-π

interactions between the stacked graphene faces, as compared to the purely physical non-bonded interactions in the control fiber sample (schematically shown in Fig. 4e). The concerted stress transfer and high strain dependence explains the excellent tensile strength of our graphene fibers assembled with aromatic amide linkages.

Lastly, we notice that our graphene fibers obtained by chemical reduction at near room temperature still exhibited a high $I_D/I_G$ ratio (Supplementary Fig. 42). Considering that FTIR and XPS spectra showed near to no trace of remnant oxygen-containing groups, the high $I_D$ can be attributed to remnant defects created during the elimination of functional groups or due to incomplete restoration of the $sp^2$ network, both of which may necessitate annealing at high temperatures of >1500 °C to repair. Thus, although connecting the individual graphene sheets with a covalent conjugating linker at sheet edges is effective in improving the assembly structure and integrated properties, the connection of graphene by edge stitching does not always lead to an ideal larger graphene sheet. This could be because of incomplete patching of the sheets due to irregular sheet edges. Similarly, certain local defects and stacking faults could not be completely avoided due to the low preparation temperature. Thus, clearly there is room for improvement, both in producing connected defect-free large graphene sheets and in obtaining perfectly stacked sheets by annihilating stacking faults through atomic diffusion during annealing. While infinitely large graphene sheets could be an ideal starting point for making graphene fibers, in practice, this is difficult to achieve. Our simple edge connection strategy, even when being far from the ideal structure, significantly influences integrated performances. Thus, our work demonstrates efficient edge linking as an alternative to replicate the ideal properties of graphene in macroscopic assemblies.

In conclusion, we developed a strategy to obtain a graphene assembly of macroscopic fibers with high mechanical strength along with excellent electrical conductivity at room temperature. Our experimental protocol involves the selective creation of amide links at the edges of graphene sheet by condensation reaction with an aromatic amine, which likely leads to the formation of extended conjugating structures. The extended conjugation in turn yields enhanced π-π interaction between the larger bridged sheets, leading to highly oriented and compact stacking of graphene into an unusual belt-shape morphology. These effects together improved the mechanical performance and electrical conductivity, and furthermore highlight the importance of controlling inter-sheet interactions in graphene assembly. Our work thus introduces a way to design high-performance macroscopic graphene fibers, which could also be interesting for the assembly of other 2D materials and for commercial industrial applications related to high-performance structural materials.

## Methods

### Materials
Expandable graphite (~300 μm) was purchased from Nanjing Xianfeng Nano Material Technology Co., Ltd. Hydrochloric acid (HCl) (~12 mol L$^{-1}$), potassium permanganate (KMnO$_4$) (≥99.5%), and sulfuric acid (H$_2$SO$_4$) (~98%) were purchased from Jiangsu Qiangsheng Functional Chemical Co., Ltd. Hydrogen peroxide (H$_2$O$_2$) (30%) was purchased from Shanghai Lingfeng Chemical Reagent Co., Ltd. Potassium persulfate (K$_2$S$_2$O$_8$) and ethanol (≥99.7%) were purchased from Sinopharm Chemical Reagent Co., Ltd. Phosphorus pentoxide (P$_2$O$_5$) was purchased from Energy Chemical Co., Ltd. Aqueous hydroiodic acid (HI, 57 wt%) was purchased from Adamas-Beta. 1,2,4,5-tetra-aminobenzene tetrahydrochloride was obtained from Sigma Aldrich. Ultrapure water was collected using a Direct-Q3 ultraviolet (UV) system.

### Synthesis of aqueous graphene oxide
Graphene oxide (GO) sheets for fiber assembly were prepared from expandable graphite, following a previously reported modified Hummers method. Owing to the presence of sulfur- or nitrogen-containing intercalation agents, the expandable graphite underwent tremendous expansion as the temperature of the system was increased up to 1000 °C for 30 s. This expansion allowed the system to reach a very high degree of oxidation during the oxidation step. Next, pre-treatment with a mixture of concentrated H$_2$SO$_4$, K$_2$S$_2$O$_8$, and P$_2$O$_5$ at 80 °C and a Hummers oxidation step using H$_2$SO$_4$ and KMnO$_4$ were carried out successively. The system was then diluted and treated with 30% H$_2$O$_2$ to reduce the residual permanganate to soluble manganese ions. After separation and thorough washing with HCl and ultrapure water, the product was collected and dispersed readily in water to produce aqueous GO.

### Fabrication of graphene fibers
We used a scalable industrial wet-spinning protocol to obtain GO fibers, followed by chemical reduction to produce high-performance graphene fibers. The GO spinning solution was extruded through a tubular needle (~160 μm inner diameter) into a coagulation bath containing 1,2,4,5-tetraaminobenzene tetrahydrochloride, mounted on a rotating plate. The needle was bent to have it parallel to the rotating direction; for large-size needles that could not be easily bent, the needle was connected to the spinning dope with a flexible hose to fix the needle direction. It should be noted that in an aromatic amine the N is less available to be bonded to the proton due to the delocalization of the lone pair of electrons into the benzene ring, which means that a substantial amount of un-ionized Ar-NH$_2$ is present in the solution, as confirmed by ninhydrin test (Supplementary Fig. 5). Furthermore, on adding GO containing carboxylic acid groups into the system, the protons have a greater tendency to protonate the oxygen on the carbonyl group, leaving more un-ionized Ar-NH$_2$.

Experimentally, we observed that solidification occurred immediately upon contact because of the amidation reaction between the amine group of the coagulant and the carboxyl groups at the graphene sheet edges. The strongly acidic environment prevents salt formation through electrostatic interaction. After 5 min of immersion in the coagulation bath and thorough washing with water, black-brown GO gel fibers were obtained. For chemical reduction, the GO fibers were suspended on parallel rods and exposed to HI vapor at 90 °C for 12 h or immersed into aqueous HI at room temperature for 12 h. The graphene fibers were then thoroughly washed alternately with water and ethanol to remove I$^-$, other soluble polyiodides (I$_3^-$ or I$_5^-$), and I$_2$. The neutral I$_2$ and other polyiodides, such as I$_3^-$ and I$_5^-$, show an intense Raman peak at 165 cm$^{-1}$ [36]. Our washed samples did not display this peak, implying that the washing procedure was successful in removing these residues.

To prepare control fibers solidified by solvent exchange, GO sheets were first dispersed in N,N-dimethylformamide (DMF) by replacing water with DMF via three successive centrifugation and washing cycles. Ethyl acetate was used as the coagulant. All other experimental conditions, including the concentration of GO spinning dope, injection rate, rotation speed of the coagulation bath, and conditions of reduction, remained identical.

### Reporting summary
Further information on research design is available in the Nature Portfolio Reporting Summary linked to this article.

## Data availability
All data that support the findings of this study are available in the main article and Supplementary Information. Source data are provided with this paper.

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

## Acknowledgements

We acknowledge financial support from the National Natural Science Foundation of China (Grant 52173288). We thank Dr. Mark Hermann Rümmeli and Yu Liu at our department for discussions. The support from the Vacuum Interconnected Nanotech Workstation (Nano-X) of Suzhou Institute of Nano-tech and Nano-bionics (SINANO), Chinese Academy of Sciences is also acknowledged.

## Author contributions

F.G. and Z.L. conceived the project and supervised experiments. L.D., T.X., J.Z., J.J., Z.S. performed materials synthesis and characterization studies. Y.X., T.L. helped with FIB cutting and data collection on STEM-in-SEM studies. Y.Z., Z.Z., R.M. conducted delamination of the amide-connected fiber and characterization of the delaminated sheets. Y.L., W.G., H.N., J.G. helped with the reaction design and mechanism analysis. Y.W. and Z.S. performed Hall-effect measurements. All authors discussed the results and F.G. organized the paper with input from all co-authors.

## Competing interests

The authors declare no competing interests.
