## [Peer Review File · Nature Communications]

Covalently Bridging Graphene Edges for Improving Mechanical and Electrical Properties of FibersThis manuscript has been previously reviewed at another journal that is not operating a transparent peer review scheme. This document only contains reviewer comments and rebuttal letters for versions considered at *Nature Communications*.

REVIEWER COMMENTS

Reviewer #2 (Remarks to the Author):

Good work. I feel so glad for the achievement of the authors in improving the mechanical and electrical properties of graphene fibers. I would like to suggest the authors pay more attention to the optimization of graphene fibers. If attention is focused on traditional carbon fibers, properties comparison between traditional carbon fibers and the achieved graphene fibers should be added, which is missing. Another one, I believe that the questions from the other two reviewers are so professional, and should be carefully addressed!

For examples, I suggest the authors weaken the edge crosslinking by the formation of amide, which causes confusion, and is even hardly convincing by the other reviewers. Instead, the optimization of graphene fibers should be stressed.

Another one is the authors weaken the claim that the amide bridging is in the in-plane direction (Comment 1,2 of Reviewer #1 and Comment 4, 5 of Reviewer #4). Guessing that if amide bridging is in the in-plane direction, the improvement of mechanical and electrical properties exactly results from the enlarged GO sheets by in-plane crosslinking adjacent GO sheets (Comment 5 of Reviewer #4). It seems that choosing large-sized GO sheets to assemble graphene fibers could simply achieve the state of the art in this manuscript.

Reviewer #4 (Remarks to the Author):

I want to thank the authors for the amount of input in addressing the reviewers' comments. They have performed many essential control experiments and very clearly prove the selectivity and effectiveness of their cross-linking protocols.

Comment 1: The authors have properly incorporated previous literature and motivated how their work is different from existing literature.

Comment 2: The authors have added more data to support the mechanical and electrical properties of their material.

Comments 4/5: The authors have properly demonstrated that their method of cross-linking is selective to the amidation of carboxylic acids. The complete absence of interlayer cross-linking however is still questionable, which is further elaborated by the questions below.

Comment 6: The authors have properly addressed the discrepancy in their C:O ratio.

Comment 7: The authors have properly added more data and elaboration on the correlation between oxidation and tensile strength.

I have a few more comment/questions:

Figure S5: Since proton exchange is extremely fast on the NMR timescale, why do different peaks arise from different combinations of amino and ammonium group instead of only the averages?

Figure S8: Why is the fiber at 0 min for IV/V flat instead of circular like for GO/I/II/III? Does the shape have an effect on the swelling directions? Moreover, could the author elaborate more on why the resistance to swelling is the result of deoxygenation? I would interpret that resistance to swelling (when GO is reacted with II and III) is the result of high degree of cross-linking, whereas slow swelling (when GO is reacted with IV and V) is the result of low degree of cross-linking. Figures: 3e/S34: Indicate in 3a explicitly which figure belongs to GO or amide-connect GO. While these results support the increased stability of the reacted GO, I do not understand how these results completely prove the absence of interlayer stacking.

Reviewer #5 (Remarks to the Author):

This paper uses a simple edge-linking protocol for the fabrication of fibers to improve their properties.

I have three questions:

1- How the mechanical properties of the fibers were tested? The authors should clarify the experiment procedure.

2- What percentage of the GO edges actually have covalent bonding as a result of this method? Can the authors provide an experimental or simulation-based estimate of the percentage of the covalent bonds formed at the edges of the GO?

3- Have the authors verified if the interlayer cross-linking has occurred?

RESPONSE TO REVIEWERS' COMMENTS

We are grateful to all the reviewers for their comprehensive review of our manuscript and the constructive comments to improve it. We have accordingly revised the text and have also provided additional experimental results to strengthen our conclusions. Below, we address each one of the comments by the reviewers. The corresponding changes to the revised text are highlighted.

Reviewer #2 (Remarks to the Author):

Good work. I feel so glad for the achievement of the authors in improving the mechanical and electrical properties of graphene fibers. I would like to suggest the authors pay more attention to the optimization of graphene fibers. If attention is focused on traditional carbon fibers, properties comparison between traditional carbon fibers and the achieved graphene fibers should be added, which is missing. Another one, I believe that the questions from the other two reviewers are so professional, and should be carefully carefully addressed!

For examples, I suggest the authors weaken the edge crosslinking by the formation of amide, which causes confusion, and is even hardly convincing by the other reviewers. Instead, the optimization of graphene fibers should be stressed.

Another one is the authors weaken the claim that the amide bridging is in the in-plane direction (Comment 1,2 of Reviewer #1 and Comment 4, 5 of Reviewer #4). Guessing that if amide bridging is in the in-plane direction, the improvement of mechanical and electrical properties exactly results from the enlarged GO sheets by in-plane crosslinking adjacent GO sheets (Comment 5 of Reviewer #4). It seems that choosing large-sized GO sheets to assemble graphene fibers could simply achieve the state of the art in this manuscript.

Response: We thank the reviewer for the positive comments. We have tried to modify the manuscript to weaken the edge cross-linking and strengthen the optimization of graphene fiber structure. Also, we have included a brief comparison with traditional carbon fibers “With the bridging at edge and improved π - π interactions between stacking sheets, the electrical conductivity of the obtained graphene fiber is more than one order of magnitude higher than that for polyacrylonitrile (PAN) based carbon fibers ($0.1-1.4 \times 10^5 \text{ S m}^{-1}$). Incidentally, this value is comparable to that of mesophase pitch (MPP)-based carbon fibers (the best value of $8.3 \times 10^5 \text{ S}$

m⁻¹), without the need for extreme high-temperature annealing.¹⁴ Although the mechanical properties are still inferior to the benchmark of carbon fibers, the simultaneous enhancement in tensile strength, modulus, and electrical conductivity of our fiber produced near room temperature highlights the importance of controlling graphene sheet assembly and the advantage of using graphene as the precursor for the fabrication of high-performance fibers.” on p15 of the revised manuscript.

We agree that the optimization of graphene fiber structure is mostly important. One consequence of the edge-linking, even in a minor percentage, can help to improve π - π stacking, which would be beneficial to optimize the fiber structure and upgrade integrated properties. Sufficiently large-sized sheets, if properly processed, are surely the best choice and starting point for producing graphene fibers, but in practice the sheets are at the most tens of micrometers large. We believe that the present strategy can to some degree overcome this limitation. We have added a discussion “While infinitely large graphene sheets could be an ideal starting point for making graphene fibers, in practice, this is difficult to achieve. Our simple edge connection strategy, even when being far from the ideal structure, significantly influences integrated performances. Thus, our work demonstrates efficient edge linking as an alternative to replicate the ideal properties of graphene in macroscopic assemblies.” On p16 in the revised manuscript.

Reviewer #4 (Remarks to the Author):

I want to thank the authors for the amount of input in addressing the reviewers’ comments. They have performed many essential control experiments and very clearly prove the selectivity and effectiveness of their cross-linking protocols.

Comment 1: The authors have properly incorporated previous literature and motivated how their work is different from existing literature.

Comment 2: The authors have added more data to support the mechanical and electrical properties of their material.

Comments 4/5: The authors have properly demonstrated that their method of cross-linking is selective to the amidation of carboxylic acids. The complete absence of interlayer cross-linking however is still questionable, which is further elaborated by the questions below.

Comment 6: The authors have properly addressed the discrepancy in their C:O ratio.

Comment 7: The authors have properly added more data and elaboration on the correlation between oxidation and tensile strength.

Response: We are happy to note that we have satisfactorily addressed all the comments and questions by the reviewer. We have addressed the unresolved issues below.

I have a few more comment/questions:

Figure S5: Since proton exchange is extremely fast on the NMR timescale, why do different peaks arise from different combinations of amino and ammonium group instead of only the averages?

Response: We appreciate the reviewer for the comments. Proton exchange is fast on the timescale of NMR, and the protons on R-NH₂ change positions so quickly, which both weakens as well as broadens the signal. If D₂O is used as the solvent, then the related signals may be barely visible, since the protons get exchanged for deuterium (R-NH₂ → R-ND₂+H₂O). We guess that 1,2,4,5-tetraaminobenzene tetrahydrochloride may to some degree oxidize to its azophenine derivatives and other decomposition products in the measurement waiting line (~ 1 day). There is a possibility that the observed peaks may reflect aryl protons (C-H) in any of the byproducts formed.

This NMR data was intended to verify the presence of -NH₂ groups, and a ninhydrin test, though qualitative, is an alternative to confirm whether amines or amino acids are present. Accordingly, **Figure S5** has been modified to describe experimental details and discussion on ninhydrin test.

Figure S5. Ninhydrin test for the coagulant solution, aqueous 1,2,4,5-tetraaminobenzene tetrahydrochloride. As aqueous 1,2,4,5-tetraaminobenzene tetrahydrochloride itself has a purple color, test with another aromatic amine 1,4-phenylenediamine dihydrochloride was also performed, and compared the results with an aliphatic amine counterpart (ethylenediamine dihydrochloride).

A few drops of ninhydrin solution were added to the test solution, and the tube was kept in a warm water bath for ca. 5 min. The test was performed in an argon-filled box. Both 1,2,4,5-tetraaminobenzene tetrahydrochloride and 1,4-phenylenediamine dihydrochloride gave a positive deep purple color after reacting with ninhydrin, confirming the presence of un-ionized Ar-NH₂ in aromatic amines, even in the presence of acid, which is related to electron delocalization into the benzene ring. In the presence of carboxyl groups, the H⁺ ions have a greater tendency to protonate the oxygen on the carbonyl group due to the more stabilized resonance structures of the intermediate. This further increases the quantity of un-ionized Ar-NH₂.

Figure S8: Why is the fiber at 0 min for IV/V flat instead of circular like for GO/I/II/III? Does the shape have an effect on the swelling directions? Moreover, could the author elaborate more on why the resistance to swelling is the result of deoxygenation? I would interpret that resistance to swelling (when GO is reacted with II and III) is the result of high degree of cross-linking,

whereas slow swelling (when GO is reacted with IV and V) is the result of low degree of cross-linking.

Response: The different shapes of the as-formed fibers are due to the different types of reactions, selective or unselective underlying their formation. To clarify this aspect, we have added an explanation in the caption of **Figure S8** “As discussed in detail below, we used different coagulants classified as, aliphatic amine in the presence of an acid (I), amine in the absence of acid (II and III), and aromatic amine in the presence of an acid (IV and V); the reactions with these coagulants of different type are quite different. The reaction with I is mainly through electrostatic interaction, but with II/III deoxygenation along with some ammonium ion intercalation takes place. The reactions with both I and II/III occur on all the negative charges on GO, resulting in a circular shape of the as-coagulated fibers. In contrast, the relatively selective reaction with groups at GO edge with coagulants IV/V triggers the formation of an unusual belt shape. These different reactions lead to large differences in the bonding between sheets, as demonstrated by their difference in stability in the presence of HCl. Swelling in all directions occurs for I, a strong resistance to swelling and negligible change of shape for II/III, and anisotropic swelling limited to the stacking directions for IV/V.”.

About possible effect of shape on the swelling direction, we guess it could be very minimal. Instead, the swelling direction and swelling degree are largely defined by the type of bond that assembles the sheets together. To exclude the possible effect of original shape, we have delicately cut the circular surface of sample GO/I into a flat sample before being subjected to swelling. The corresponding results and discussions have been included in **Figure S9** in the revised SI.

Figure S9. Stability of the fiber with coagulant I under the same condition as in Figure S8, except that the circular shape was cut from the top and left-side into a flat piece before adding HCl to initiate swelling. We also tried to cut the solvent-exchanged GO sample in a similar way, but this was not possible because the sample easily broke into several small parts, probably due to the weak binding of sheets.

With swelling and time, the flat morphology gradually evolved into a round shape, largely to minimize surface energy. This result shows that the original shape of the fiber does not play a determining role in the swelling behavior, which is instead mainly determined by the bonds that assemble the sheets. In the absence of a chemical bond to restrict swelling (I), the fiber will evolve with time into a circular shape so as to minimize its surface energy; if swelling is restricted in all directions (II and III), no swelling would be observed. If the restriction is present only in certain direction, then anisotropic swelling occurs.

We would also like to thank the reviewer for this suggestion on data interpretation. Our original thought was that deoxygenation may bring about swelling resistance in aqueous HCl through expelling water (confirmed by contact angle test). We agree with the reviewer that the cross-linking degree also plays a role and should be accounted for. In II and III, the resistance to swelling and the almost negligible change of shape, along with the circular shape, indicates a high degree of cross-linking in both lateral and interlayer directions. In IV and V, the negligible swelling in the lateral directions implies high-degree of chemical cross-linking, while the slow swelling in the stacking direction may suggest improved π - π interaction or some low-degree chemical cross-linking. The anisotropic swelling phenomenon further confirms the selective chemical cross-linking in our strategy. We have accordingly modified the discussions in the caption of **Figure S8** and also on **p12** of the revised manuscript.

Figures: 3e/S34: Indicate in 3a explicitly which figure belongs to GO or amide-connect GO. While these results support the increased stability of the reacted GO, I do not understand how these results completely prove the absence of interlayer stacking.

Response: Yes, Figures 3e and S34 show the increased stability and increase in lateral size of delaminated sheets. The improved stability could come from the improved π - π interactions. As NMP can only disrupt π - π interaction and cannot break chemical bonds, the successful delamination is evidence that no strong chemical interactions exist. This is in contrast to the fibers coagulated with II/III in Figure S8, which cannot be swollen and dissociated in NMP. At the same time, the obviously enlarged size after delamination confirms the presence of linking between the edges. To clarify these points, we have marked the samples in Figure 3, and also included the following explanation “The successful delamination confirms that weak interactions are present in the vertical stacking direction, which are mostly π - π or minimal chemical interactions. The excellent stability in the lateral direction and the obviously enlarged sheet size even after mechanical shaking indicates the presence of strong chemical interactions at edges. Taken together, these results, including the selective swelling limited to the stacking direction either in water, acid or NMP, the delamination of interlayer connections, the enlarged sheet size after delamination, along with the almost no change in the in-plane surface structure, substantiate the conclusion that the covalent amide linkage bridging is basically at the edges. In the vertical stacking direction, enhanced π - π interactions/very low-degree of chemical cross-linking is present.” on p12 of the revised manuscript.

Reviewer #5 (Remarks to the Author):

This paper uses a simple edge-linking protocol for the fabrication of fibers to improve their properties.

I have three questions:

1- How the mechanical properties of the fibers were tested? The authors should clarify the experiment procedure.

Response: We thank the reviewer for the comments. We have supplemented more details on mechanical property measurement in the experimental section.

2- What percentage of the GO edges actually have covalent bonding as a result of this method? Can the authors provide an experimental or simulation-based estimate of the percentage of the covalent bonds formed at the edges of the GO?

Response: Even since we started this work nearly 4 years ago, we have been trying to give a quantitative estimate of the percentage of the formed amide linking on the edge. However, due to the low amount of carboxyl as compared to total carbon and the complexity of sheet edge conditions, it has been difficult to precisely estimate either by experiment or simulation. It might be possible using an isotope method.

3- Have the authors verified if the interlayer cross-linking has occurred?

Response: We have excluded significant interlayer chemical linking by investigating i) change of surface structure, ii) the swelling stability, and iii) the delamination behavior. We have added more discussions on this aspect in the revised version (please see response to comment 3 of Reviewer 4#).

REVIEWER COMMENTS

Reviewer #4 (Remarks to the Author):

I would like to express my gratitude to the authors for their substantial effort in addressing the reviewers' comments. As a final comment, I would really appreciate if the data concerning swelling (Fig. 3d/S8d) can be expressed quantitatively, ideally accompanied with error bars.

Concerning point 2 below, I imagine that using a FTIR technique was challenging.

2- What percentage of the GO edges actually have covalent bonding as a result of this method? Can the authors provide an experimental or simulation-based estimate of the percentage of the covalent bonds formed at the edges of the GO?

Response: Even since we started this work nearly 4 years ago, we have been trying to give a quantitative estimate of the percentage of the formed amide linking on the edge. However, due to the low amount of carboxyl as compared to total carbon and the complexity of sheet edge conditions, it has been difficult to precisely estimate either by experiment or simulation. It might be possible using an isotope method.

Reviewer #5 (Remarks to the Author):

The authors have addressed my questions and I can recommend their paper for publication.

RESPONSE TO REVIEWERS' COMMENTS

We are grateful to all the reviewers for their comprehensive review of our manuscript and the constructive comments to improve it. We have accordingly revised the text. Below, we address each one of the comments by the reviewers.

Reviewer #4 (Remarks to the Author):

I would like to express my gratitude to the authors for their substantial effort in addressing the reviewers' comments. As a final comment, I would really appreciate if the data concerning swelling (Fig. 3d/S8d) can be expressed quantitatively, ideally accompanied with error bars.

Response: We thank the reviewer for the comment. We have measured the swelling at 15 locations for each of the samples, and the corresponding values along with the error bars are given in **Fig. 3d/S8d**. The caption has also been correspondingly modified. We have also included error bars for the data in **Figure S9, S13, S17, S18, S19, S20, and S34**.

Revised Figure 3d. (d) Optical images of the amide-connected GO fiber swelling in NMP in the axial and stacking directions; right panel: XRD patterns of the fibers. The anisotropic swelling and the longer time confirm the presence of selective amide linkages at edges and enhanced π - π interactions. Error bars correspond to the statistical error from independent measurements from at least 15 locations.

Revised Figure S8d. (d) stability in HCl. Experimental conditions were the same for all the amines and the amino group concentration was 0.02 M in all cases. Aqueous HCl was selected because water can disrupt the π - π interaction due to the hydrophilic surface of GO, and acid can break the ion bridge formed by electrostatic interactions. Error bars were estimated from statistical analysis of measurements on 15 locations on each sample.

Concerning point 2 below, I imagine that using a FTIR technique was challenging.

2- What percentage of the GO edges actually have covalent bonding as a result of this method? Can the authors provide an experimental or simulation-based estimate of the percentage of the covalent bonds formed at the edges of the GO?

Response: Even since we started this work nearly 4 years ago, we have been trying to give a quantitative estimate of the percentage of the formed amide linking on the edge. However, due to the low amount of carboxyl as compared to total carbon and the complexity of sheet edge conditions, it has been difficult to precisely estimate either by experiment or simulation. It might be possible using an isotope method.

Response: We thank the reviewer for the comment. Indeed, we found that determining the extent of covalent bonding using FT-IR technique is challenging. Assuming that there are two types of bonds, covalent ($\text{GO-COOH} + \text{H}_2\text{N-R} \rightarrow \text{GO-CON-NHR}$; amide) and ionic ($\text{GO-COOH} + \text{H}_2\text{N-R} \rightarrow \text{GO-COO} + \text{H}_3\text{N-R}$; carboxylate + ammonium), perhaps we can compare the intensities of carbonyl stretching at frequencies corresponding to the amide ($\sim 1650 \text{ cm}^{-1}$) versus that due to carboxylate (~ 1580 and $\sim 1360 \text{ cm}^{-1}$). However, in practice, these signals are not resolved sufficiently clearly and other assumptions may be required (e.g., all of the functional groups react, no side reactions, etc.).

We have also tried to use XPS to give a quasi-quantitative percentage of amide formation. We planned to use carboxyl percentage from XPS C 1s and edge carbon percentage from simple geometrical estimation to calculate the percentage of carboxyl at the edge sites. In this manner, with the percentage of transformation of the carboxyl to amide from XPS, we then can estimate the percentage of edge linking. However, the carboxyl percentage from XPS is much greater than the percentage of edge carbons in a graphene sheet of $\sim 30 \mu\text{m}$ obtained by simple geometrical calculation (these values are $\sim 2.8\%$ and 0.023% , respectively). We have considered the different reasons for the unexpected discrepancies in these percentages. The most probable explanation could be due to the irregular geometry of the GO sheets. In estimating the edge carbon sites, we assumed that the GO sheets adopt a defined square or round shape and size. However, if these sheets are irregular or in another case have large length to width ratio, then the perimeters may be significantly longer than assumed. This could complicate any determinations of edge carboxyl percentage and transformation yield.

Thus, determining the extent of edge bonding is expected to be hard, if not impossible, and will probably require significant assumptions regardless of the method used.

Reviewer #5 (Remarks to the Author):

The authors have addressed my questions and I can recommend their paper for publication.

Response: We thank the reviewer for the recommendation of the paper publication.

REVIEWERS' COMMENTS

Reviewer #4 (Remarks to the Author):

Thank you for your answers. I do not have further comments on the manuscript which can now be published.